# Dietary Micronutrient Status and Relation between Micronutrient Intakes and Overweight and Obesity among Non-Pregnant and Non-Lactating Women Aged 18 to 49 in China

**DOI:** 10.3390/nu14091895

**Published:** 2022-04-30

**Authors:** Lahong Ju, Xiaoqi Wei, Dongmei Yu, Hongyun Fang, Xue Cheng, Wei Piao, Qiya Guo, Xiaoli Xu, Shujuan Li, Shuya Cai, Liyun Zhao

**Affiliations:** Chinese Center for Disease Control and Prevention, Laboratory of Trace Element Nutrition of National Health Commission, National Institute for Nutrition and Health, Beijing 100050, China; julh@ninh.chinacdc.cn (L.J.); xq437073568@163.com (X.W.); yudm@ninh.chinacdc.cn (D.Y.); fanghy@ninh.chinacdc.cn (H.F.); chengxue@ninh.chinacdc.cn (X.C.); piaowei@ninh.chinacdc.cn (W.P.); guoqy@ninh.chinacdc.cn (Q.G.); xuxl@ninh.chinacdc.cn (X.X.); lisj@ninh.chinacdc.cn (S.L.); caisy@ninh.chinacdc.cn (S.C.)

**Keywords:** micronutrient intakes, overweight, obesity, women, non-pregnant, non-lactating

## Abstract

Women between the ages of 18 and 49 are women of reproductive age, for whom physical health and nutritional status are closely related to successful pregnancy, good pregnancy outcomes and the nurturing of the next generation. Overweight and obesity have become important nutrition and health problems of women aged 18–49 years in China. In social life, non-pregnant and non-lactating Chinese women aged 18–49 are the most vulnerable and neglected group. At present, there are no research data on their dietary micronutrient intake, and the relationship between dietary micronutrient intake and overweight and obesity in China. However, non-pregnant and non-lactating women aged 18–49 are the best window of opportunity to implement strategies, correct nutrition and improve physical health. It remains to be explored whether their overweight and obesity are related to inadequate dietary micronutrient intake. The aim of this study was to evaluate dietary micronutrient intake in non-pregnant and non-lactating Chinese women aged 18–49 years, and to analyze the relationship between dietary micronutrient intake and overweight and obesity. Data were obtained from 2015 China Adult Chronic Disease and Nutrition Surveillance (CACDNS 2015). In CACDNS 2015, 12,872 women aged 18 to 49 years (excluding pregnant women and lactating mothers) were surveyed for a three-day 24 h dietary recall and a three-day household weighing of edible oil and condiments. The average daily dietary intake of micronutrients was calculated according to the Chinese food composition table. In 2015, the median intake of vitamin A, vitamin B_1_, vitamin B_2_, vitamin C and folate in non-pregnant and non-lactating women aged 18–49 years in China was 267.0 μg RE/day, 0.7 mg/day, 0.6 mg/day, 63.5 mg/day and 121.0 μg/day, respectively. The median mean intake of vitamin A, niacin, calcium and zinc in overweight/obese group was lower than that in non-overweight/obese group, and the difference was statistically significant (*p* < 0.05). Multivariate Logistic regression analysis showed that vitamin A intake (Q3 vs. Q1: OR = 0.785, 95% CI: 0.702~0.878; Q4 vs. Q1: OR = 0.766, 95% CI: 0.679~0.865), niacin intake (Q2 vs. Q1: OR = 0.801, 95% CI: 0.715–0.898; Q3 vs. Q1: OR = 0.632, 95% CI: 0.554~0.721; Q4 vs. Q1: OR = 0.662, 95% CI: 0.568~0.772), Zinc intake (Q4 vs. Q1: OR = 0.786, 95% CI: 0.662~0.932) were a protective factor for overweight/obesity in women, while vitamin B_2_ intake (Q2 vs. Q1: OR = 1.256, 95% CI: 1.120~1.408; Q3 vs. Q1: OR = 1.416, 95% CI: 1.240~1.617; Q4 vs. Q1: OR = 1.515, 95% CI: 1.293–1.776), vitamin E intake (Q2 vs. Q1: OR = 1.114, 95% CI: 1.006–0.235; Q3 vs. Q1: OR = 1.162, 95% CI: 1.048~0.288; Q4 vs. Q1: OR = 1.234, 95% CI: 1.112–1.371) was a risk factor for overweight/obesity in females. The intakes of most dietary micronutrients in non-pregnant and non-lactating women aged 18–49 in China were low. The intakes of dietary vitamin A, niacin and zinc were negatively correlated with the risk of overweight/obesity, while the intakes of vitamin B_2_ and vitamin E were positively correlated with the risk of overweight/obesity.

## 1. Introduction

Women between the ages of 18 and 49 are women of reproductive age, for whom physical health and nutritional status are closely related to successful pregnancy, good pregnancy outcomes, and nurturing of the next generation. At the same time, it also affects their own labor productivity, and is closely related to the occurrence of nutritional diseases such as osteoporosis and iron deficiency in middle-aged and elderly women in the future [1,2]. The China Nutrition and Health Surveillance of Chinese residents also showed [3] that the overweight (body mass index (BMI) ≥ 24 kg/m^2^) and obesity (body mass index (BMI) ≥ 28 kg/m^2^) rates of Chinese non-pregnant and non-lactating women aged 18–44 increased from 21.8% and 6.1% in 2002 to 24.9% and 8.8% in 2012, respectively, and those of women aged 45–59 increased from 31.4% and 12.9% in 2002 to 38.3% and 15.8% in 2012, respectively. The results suggest that over one-third of Chinese women aged 18–49 are overweight or obese, and showed an increasing trend during the past decade. Multiple studies have highlighted the detrimental role of overweight/obesity in cancer, with almost 55% of cancers diagnosed in women being considered to be overweight- and obesity-related cancers. Women who are overweight and obese have an increased risk of endometrial and breast cancer [4,5,6]. Therefore, overweight and obesity have become important nutrition and health problems in women aged 18–49 in China, and have become one of the public health problems that need to be solved urgently.

Obesity is a major risk factor for chronic non-communicable diseases such as hypertension, diabetes, cardiovascular disease and cancer, and has become a serious problem threatening global health [7]. Obesity is a long-term nutritional disorder in which energy intake exceeds energy expenditure. Previous studies have shown that micronutrients (including vitamins and minerals) play important roles in energy and blood lipid metabolism [8,9,10,11] through several mechanisms [12,13,14,15]. For instance, several minerals and vitamins could promote the expressions of UCP1–3 mRNA and improve mitochondrial function. The changes in these indexes can upregulate thermogenesis, promote lipolysis and increase energy consumption. Moreover, Micronutrients may play a role as in the synthesis and regulation of neurotransmitters in the central nervous system for influencing the control of food intake. A Studies have found that Vitamins and minerals seem to have an appetite-related effect for women. vitamin and mineral supplements could attenuate appetite in women, the attenuate in appetite that often accompanies body-weight loss [16].

Previous studies have also found that there is a generally deficient intake of vitamin A, vitamin B_2_, vitamin C, vitamin D, vitamin E, niacin, calcium, iron and zinc among obese people [16,17,18,19,20,21,22,23,24,25]. This may be related to the fact that the deficiency of micronutrients can cause fatigue, reduced exercise and energy consumption of the human body, and then lead to obesity [26]. Moreover, Li, Y. conducted a randomized, double-blind, placebo-controlled intervention trial on 96 obese women (body mass index (BMI) ≥ 28 kg/m^2^) aged 18–55 years in China. The 29-ingredient multivitamin and mineral supplementation over 26 weeks could reduce the weight and obesity of them [27], which verified the regulatory effect of micronutrients on obesity.

National Nutrition Plan (2017–2030) was issued by The State Council General Office of China in 2017 [28] lists folic acid deficiency rate, anemia rate among pregnant women, etc., as China’s nutrition development goals, carries out nutritional evaluation and dietary guidance for women before pregnancy, during pregnancy and after delivery, and actively guides women in perinatal period to supplement micronutrients containing folic acid, iron, etc. In social life, non-pregnant and non-lactating Chinese women aged 18–49 are the most vulnerable and neglected group. At present, there are no research data on their dietary micronutrient intake, and the relationship between dietary micronutrient intake and overweight and obesity in China. Micro-nutrient deficiencies in non-pregnant and non-lactating women aged 18–49 years can adversely affect fertility, pregnancy outcomes and the risk of congenital disability, harming the health of mother and offspring [29]. However, non-pregnant and non-lactating women aged 18–49 are the best window of opportunity to implement strategies, correct nutrition and improve physical health. China Nutrition and Health Surveillance (2015–2017) [30] was the latest cross-sectional survey and was nationally representative. This study aims to use the 2015 China Adult Chronic Disease and Nutrition Surveillance (CACDNS 2015) data to evaluate 18- to 49-year-old non-pregnant non-lactating women dietary micronutrient intake status. The relationship between dietary micronutrient intake and overweight and obesity was analyzed to provide scientific evidence for the scientific diet and health promotion of non-pregnant and non-lactating women aged 18–49 in China.

## 2. Materials and Methods

### 2.1. Study Design and Samples

The data were obtained from the data of China Adult Chronic Disease and Nutrition Surveillance in 2015 (CACDNS 2015), which was organized by the National Institute of Nutrition and Health, Chinese Center for Disease Control and Prevention, based on a major public health project organized by CDC of the Health Commission of the People’s Republic of China. Multistage stratified random sampling method was used to select 302 monitoring sites in 31 provinces of China. At each monitoring site, multistage stratified random sampling method was used to select permanent residents aged 18 and above (or non-local residents who had resided for more than 6 months) for a questionnaire survey on chronic diseases and nutrition. The survey includes four parts: inquiring survey, anthropometric measurements, laboratory test and dietary survey. The survey requires that at least 270 households and at least 612 permanent residents aged 18 and above be investigated at each monitoring site [30]. After data cleaning, a total of 12,872 women aged 18–49 were included in this study (excluding pregnant women, lactating mothers and other special groups). The project was reviewed by the Ethics Review Committee of the Chinese Center for Disease Control and Prevention (Batch No. 201519-B, 2015). All the subjects signed informed consent before the investigation.

### 2.2. Data Collection and Measurements

The questionnaire was designed by the National Project team, including the registration tables for basic information of family members, 3-day weight of household cooking oil and condiments, 3-day number of family-cooked dinners, and a 24 h dietary recall table. The age, sex, region, education level and marital status of the respondents were collected by the personal social demographic survey questionnaire. The uniformly trained investigators collected the information of the respondents through face-to-face inquiry, and collected the 24 h dietary recall information for three consecutive days (two week days and one weekend day) [31], including all foods eaten at home and out, and kept a detailed record of the names of all foods consumed and the amount of food consumed. At the same time, food weight record [32] was used to record the consumption of household edible oil and condiments during the survey, and food scales were used to record the intake of various edible oil and salt and other major condiments for three consecutive days. Meanwhile, the number of diners was recorded. The survey time was synchronized with the survey time of 24 h 3-day dietary recall.

There was recall bias in 24 h dietary recall. In the present study, the investigators issued a personal meal record form to each of the respondents in advance, and the respondents recorded their meals on the same day, including all foods eaten at home and out, and kept a detailed recorded of the names of all foods consumed and the amount of food consumed, which was asked for and verified and recorded on the 24 h dietary questionnaire by the investigator the next day.

The anthropometric measurements included indicators such as height and weight. Height and weight values were measured centrally by uniformly trained investigators using standard methods [33] and certified and approved measuring instruments (TZG height altimeter, Tanita HD-390 electronic weight scale) designated by the National Project Team, with readings accurate to 0.1 cm and 0.1 Kg, respectively. When the height and weight were measured, the subjects were required to take off their shoes and hats and wear lightweight clothes for one time.

### 2.3. Dietary Assessment

Household edible oil and condiment intakes were divided by individual intake according to the amount of family meals and the proportion of individual energy of family members. Personal food consumption included food, cooking oil and condiment consumption for a 24 h 3-day dietary recall. The average daily dietary intake of micronutrients was calculated according to the Chinese food Composition table [34,35,36].

### 2.4. Assessment of BMI

Body mass index (BMI) was calculated as weight (kg)/height^2^ (m^2^), referring to Guidelines for prevention and Control of Overweight and Obesity in Chinese Adults [37], BMI < 18.5 kg/m^2^ is underweight, 18.5 kg/m^2^ ≤ BMI < 24 kg/m^2^ is normal, 24 kg/m^2^ ≤ BMI < 28 kg/m^2^ is overweight, BMI ≥ 28 kg/m^2^ is obesity. In this study, BMI < 24.0 kg/m^2^ was divided into non-overweight/obese group, BMI > 24.0 kg/m^2^ was overweight/obese group.

### 2.5. Covariates

A wide range of potential confounders were accounted for: (1) age was categorized as 18–29 years, 30–39 years, 40–49 years; (2) area type was categorized as urban and rural; (3) education level was categorized as low (junior high and below), moderate (high school/technical secondary school/technical school) or high (junior college and above); (4) marital status was categorized as, married/cohabitation and other (including widowed/divorced/separated/unmarried; (5) occupations were classified into occupational people and non-occupational (include school students, retired/unemployed).

### 2.6. Statistical Analysis

SAS 9.4 software (SAS Institute Inc., Cary, NC, USA) was used for data cleaning and statistical analysis. Demographic characteristics were expressed as frequency and percentage, and comparisons between different groups were performed using chi-square test. Micronutrient intake was measured by median and interquartile spacing (IQR), and Wilcoxon test was used for comparison between groups. Multivariate analysis was performed by Logistic regression analysis, *p* < 0.05 was considered statistically significant.

## 3. Results

### 3.1. General Characteristics of the Participants

In 2015, a total of 12,872 non-pregnant and non-lactating women aged 18–49 were surveyed, among whom 2517 (19.6%) were aged 18–29, 3580 (27.8%) were aged 30–39, and 6775 (52.6%) were aged 40–49, 5296 (41.1%) in urban areas, and 7576 (58.9%) in rural areas. Female with junior high school degree or below and married or cohabiting were the main groups, accounting for 73.8% (9504) and 94.0% (12,098). A total of 5634 (43.8%) of the surveyed women aged 18–49 in China were overweight and obese, accounting for 43.8%. The differences of overweight and obesity rate in age, urban and rural areas, educational level, marital status, and occupation were statistically significant, and detailed results are shown in Table 1.

### 3.2. Micronutrient Intake in Non-Pregnant and Non-Lactating Women Aged 18–49

Table 2 shows dietary vitamin and mineral intake in non-pregnant and non-lactating women aged 18–49 years. The median intakes of vitamin A, vitamin B_1_, vitamin B_2_, niacin, vitamin C, vitamin E and folate in Chinese women aged 18–49 years were 267.0 μg RE/day, 0.7 mg/day, 0.6 mg/day, 11.9 mg NE/day, 63.5 mg/day, 25.1 mg/day and 121.0 μg/day, respectively. The median intakes of calcium, iron, zinc, potassium, sodium and selenium were 275.6 mg/day, 16.8 mg/day, 8.5 mg/day, 1298.7 mg/day, 5448.8 mg/day and 32.6 μg/day, respectively. There were statistically significant differences in the average intake of vitamin A, vitamin B_2_, vitamin E, niacin, calcium, zinc and sodium between overweight/obese group and non-overweight/obese group (*p* < 0.05).

### 3.3. Dietary Univariate Analysis of Overweight and Obesity in Non-Pregnant and Non-Lactating Women Aged 18–49 Years

According to the 25th, 50th and 75th percentile of average dietary micronutrient intake of non-pregnant and non-lactating women aged 18–49, they were divided into four levels of Q1, Q2, Q3 and Q4 for univariate analysis. Vitamin A, vitamin B_2_, niacin, Vitamin E, zinc and sodium intake levels were associated with overweight/obesity, and the differences were statistically significant (*p* < 0.05), as shown in Table 3 and Table 4.

### 3.4. Multifactorial Analysis of Overweight and Obesity in Non-Pregnant and Non-Lactating Women Aged 18–49 Years

Table 5 shows the multi-factor analysis of overweight and obesity in non-pregnant and non-lactating women aged 18–49. Overweight and obesity in non-pregnant and non-lactating women aged 18–49 was taken as the dependent variable (yes = 1, no = 0). According to the 25th, 50th and 75th percentile of dietary micronutrient intake, dietary micronutrient intake was divided into Q1, Q2, Q3 and Q4 levels, each intake level of Q1 was taken as the reference group, and statistically significant variables in univariate analysis were included in the stepwise method for unconditional logistic regression to analyze the impact of dietary factors intake level on overweight and obesity in women aged 18–49. The results showed that after adjusting for age, area type, marital status, education level and occupation level, when vitamin A intake levels were at Q3 and Q4 levels, that is, the risk of overweight and obesity decreased when vitamin A intake was 267.00~485.67 μgRAE/day and ≥485.67 μgRAE/day (OR = 0.785, 95% CI: 0.702~0.878; OR = 0.766, 95% CI: 0.679~0.865). When vitamin B_2_ intake levels were at Q2, Q3 and Q4 levels, that is, when vitamin B_2_ intake were 0.46~0.60 mg/day, 0.60~0.77 mg/day, and ≥0.77 mg/day, the risk of overweight and obesity increased (OR = 1.256, 95% CI: 1.120~1.408; OR = 1.416, 95% CI: 1.240~1.617; OR = 1.515, 95% CI: 1.293~1.776). When niacin intake levels were at Q2, Q3 and Q4 levels, that is, when niacin intake levels were 8.72~11.88 mg NE/day, 11.88~15.73 mg NE/day and 15.73 mg NE/day, the risk of overweight and obesity decreased (OR = 0.801, 95% CI: 0.715–0.898; OR = 0.632, 95% CI: 0.554~0.721; OR = 0.662, 95% CI: 0.568~0.772). When vitamin E intake levels were at Q2, Q3 and Q4, that is, when vitamin E intake was 16.48~25.12 mg/day, 25.12~37.99 mg/day and 37.99 mg/day, the risk of overweight and obesity increased (OR = 1.114, 95% CI: 1.006–0.235; OR = 1.162, 95% CI: 1.048–0.288; OR = 1.234, 95% CI: 1.112–1.371). Zinc intake at Q4 level (≥10.56 mg/day) was a protective factor for overweight and obesity (OR = 0.786, 95% CI: 0.662~0.932).

## 4. Discussion

In 2015, the median intake of vitamin A, B_1_, B_2_, C and folate in non-pregnant and non-lactating women aged 18–49 in China was 267.0 μg RE/day, 0.7 mg/day, 0.6 mg/day, 63.5 mg/day, 121.0 μg/day, respectively, which were lower than the recommended dietary vitamin intake (RNI) of Chinese residents. The mean median intake of niacin 11.9 mg NE/day reached RNI, and the mean intake of vitamin E exceeded about 1.5 times that of RNI [38]. The average median intakes of calcium, iron and selenium were 275.6 mg/day, 16.8 mg/day and 32.6 μg/day, respectively, which were lower than RNI. The average median intake of zinc was 8.5 mg/day, which reached RNI. The median average intake of potassium was 1298.7 mg/day, lower than the adequate intake (AI), and the median average intake of sodium was 5448.8 mg/day, which was 3.5 times higher than AI [38]. Compared with the micronutrient intake of women aged 20 and above in the United States [39] and Japan [40], the intake of vitamin A, vitamin B_1_, vitamin B_2_, vitamin C and calcium in non-pregnant and non-lactating women aged 18–49 in China was lower, iron and sodium intake were higher, zinc intake was higher than in Japan and lower than in the United States. It is suggested that the intakes of niacin and zinc in non-pregnant and non-lactating women aged 18–49 are basically adequate, while the intakes of vitamin E and sodium are higher and the intakes of other micronutrients are inadequate.

The study also found that the intake of vitamin A, niacin, calcium, and zinc was lower in the overweight/obese group than in the non-overweight/obese group, while the intake of vitamin E and sodium was higher in the non-overweight/obese group, which was consistent with several relevant research results [22,23,24,25,41,42].

Multivariate logistic regression analysis showed that after adjusting for age, area types, marital status, education level and employment level, vitamin A intake Q3, Q4 level, niacin intake Q2, Q3, Q4 level, and zinc intake Q4 level were protective factors of overweight and obesity in women compared with Q1 level, which is consistent with several relevant research results [22,23,24,25,41,42]. This may be related to the fact that vitamin A is a fat-soluble vitamin necessary for maintaining normal metabolism and function of human body, participates in the regulatory function of body fat, plays a key role in fat formation and lipid accumulation, and can reduce visceral fat, subcutaneous fat, and total fat [22,43,44]. Insufficient niacin can increase the production of active oxidants, enhance inflammatory response, and promote fat production [45]. In addition, niacin can control appetite and lead to obesity by affecting brain neurotransmitter metabolism, thus affecting eating behavior [46]. Some studies have found that zinc may be the medium of leptin synthesis, and zinc deficiency can reduce the secretion of leptin. Leptin can reduce appetite, improve energy metabolism efficiency, reduce fat and weight [47,48]. Zinc deficiency not only leads to the decrease of leptin, but also leads to unhealthy eating behaviors such as taste disorder, partial eclipse, and different diet, which leads to obesity [49]. The NHANES (2007–2014) also found that dietary zinc intake was negatively correlated with BMI and waist circumference (WC) in women, and the correlation increased with the increase of risk level. The influence of zinc intake on BMI and WC was greater in overweight and obese people [40], but no such association was found in men [42,50]. Micronutrients mainly come from plant foods such as milk, cereal and potato, and vegetables [44,51]. Therefore, non-pregnant and non-lactating overweight and obese women aged 18–49 in China are recommended to increase their intake of plant-based foods, especially animal offal, fish and shrimp, egg yolk, dairy products and whole grains, which are rich in vitamin A, niacin and zinc.

Multivariate logistic regression analysis in this study also showed that compared with Q1 level, vitamin B_2_ intake Q2, Q3, Q4 level and vitamin E intake Q2, Q3, Q4 level were risk factors for overweight and obesity in women. In several studies, riboflavin intake was negatively correlated with BMI [52,53], contrary to the results of this study. Shi-sheng Zhou’s time-lag regression model analysis found that the prevalence of obesity increased with the increase of riboflavin intake, and riboflavin intake had a delayed impact on the prevalence of obesity [54], which may be the reason for the inconsistent results of this study and other studies. In different studies, vitamin E has a negative or positive relationship with obesity or no effect [53,55,56], and some studies have found that vitamin E intake level is negatively correlated with overweight and obesity in women, but not in men [57]. This study showed that vitamin E intake was positively correlated with the risk of overweight and obesity. This may be related to the fact that vitamin E exhibits cholesterol-lowering effects [58,59]. Vitamin E prevents cholesterol accumulation, which leads to obesity through amplification of inflammatory responses [60] and increasing insulin resistance [61]. Moreover, it is also related to the fact that vitamin E mainly comes from cooking oil in Chinese residents. The percentage of vitamin E from cooking oil was 62.4% in non-pregnant and non-lactating women aged 18 to 49 in China (see Figure A1). According to the Report Chinese Residents’ Nutrition and Chronic Diseases (2020) [62], the intake of cooking oil is continuing to rise, and excessive intake of cooking oil will increase the risk of obesity [63]. Therefore, non-pregnant and non-lactating overweight and obese women aged 18–49 in China are suggested to reduce their intake of cooking oil.

The results of this study have at least three limitations. First, the amount of micronutrient intake from nutrient supplements and fortified foods was not included in the study, and the overall levels of micronutrient intake were somewhat underestimated. Second, the cross-sectional data in this study failed to establish a causal relationship between dietary micronutrients and overweight and obesity. Third, this study did not include the impact of physical activity on overweight and obesity, so there are certain limitations. Future prospective studies and physical activity variables will be added to explore the impact of dietary factors on overweight and obesity.

## 5. Conclusions

The intake of most dietary micronutrients in non-pregnant and non-lactating women aged 18–49 in China is low. The intake of dietary vitamins A, niacin and zinc was negatively correlated with the risk of overweight and obesity, while the intake of vitamins B_2_ and E was positively correlated with the direction of overweight and obesity. Non-pregnant and non-lactating overweight and obese women aged 18–49 in China are recommended to reduce their intake of cooking oil, and increase their intake of plant-based foods, especially animal offal, fish and shrimp, egg yolk, dairy products and whole grains, which are rich in vitamin A, niacin and zinc.

## Figures and Tables

**Table 1 nutrients-14-01895-t001:** Basic information of non-pregnant and non-lactating females aged 18–49 years.

Variable	Overweight and Obese				
	*n* (%)	No	Yes	X^2^	*p*
Total	12,872 (100)	7238 (56.2)	5634 (43.8)		
Age group, years				496.674	<0.0001
18–29	2517 (19.6)	1805 (71.7)	712 (28.3)		
30–39	3580 (27.8)	2211 (61.8)	1369 (38.2)		
40–49	6775 (52.6)	3222 (47.6)	3553 (52.4)		
Area type				11.281	0.0008
Urban	5296 (41.1)	3071 (58.0)	2225 (42.0)		
Rural	7576 (58.9)	4167 (55.0)	3409 (45.0)		
Education level				184.120	<0.0001
Low	9504 (73.8)	5026 (52.9)	4478 (47.1)		
Moderate	1654 (12.8)	1024 (61.9)	630 (38.1)		
High	1714 (13.3)	1188 (69.3)	526 (30.7)		
Marital status				51.234	<0.0001
Married/cohabitation	12098 (94.0)	6707 (55.4)	5391 (44.6)		
Other	774 (6.0)	531 (68.6)	243 (31.4)		
Employment				6.910	0.0086
Non-occupational	3093 (24.0)	1676 (54.2)	1417 (45.8)		
Occupational	9779 (76.0)	5562 (56.9)	4217 (43.1)		

**Table 2 nutrients-14-01895-t002:** Median intake of major vitamins and minerals in non-pregnant and non-lactating women aged 18–49 years (g/day) [M (P25, P75)].

Variable	Total	Non-Overweight/Obese	Overweight/Obese
	M (P25, P75)	M (P25, P75)	M (P25, P75)
Vitamin A (μg RE/day)	267.0 (141.8, 485.7)	285.8 (150.0, 505.4)	246.1 (132.1, 457.6) ^
Vitamin B_1_ (mg/day)	0.7 (0.5, 0.9)	0.7 (0.5, 0.9)	0.7 (0.5, 0.9)
Vitamin B_2_ (mg/day)	0.6 (0.5, 0.8)	0.6 (0.5, 0.8)	0.6 (0.5, 0.8) ^
Niacin (mg NE/day)	11.9 (8.7, 15.7)	12.3 (9.1, 16.0)	11.3 (8.3, 15.3) ^
Vitamin C (mg/day)	63.5 (40.5, 94.3)	64.0 (40.7, 95.0)	62.9 (40.2, 93.6)
Vitamin E (mg/day)	25.1 (16.5, 38)	24.4 (15.9, 36.9)	26.0 (17.2, 39.3) ^
Folate (μg/day)	121 (80.9, 174.1)	121.8 (81.6, 175.2)	119.8 (80.4, 172.1)
Calcium (mg/day)	275.6 (203.2, 378.3)	278.0 (205.0, 382.8)	272.3 (201.5, 371.9) *
Iron (mg/day)	16.8 (13.6, 21.3)	16.8 (13.6, 21.5)	16.8 (13.6, 21.2)
Zinc (mg/day)	8.5 (6.8, 10.6)	8.6 (6.9, 10.7)	8.3 (6.6, 10.3) ^
Potassium (mg/day)	1298.7 (1025.4, 1649)	1296.2 (1024.3, 1655.5)	1300.2 (1026.9, 1641.2)
Sodium (mg/day)	5448.8 (3930.5, 7433.9)	5371.3 (3886.1, 7323.0)	5559.0 (3996.1, 7528.3) ^
Selenium (μg/day)	32.6 (23.8, 44.5)	32.6 (23.7, 44.6)	32.6 (23.9, 44.4)

* *p* < 0.05 compared with non-overweight/obese group, ^ *p* < 0.01 compared with non-overweight/obese group.

**Table 3 nutrients-14-01895-t003:** Univariate analysis of the effect of dietary vitamin intake on overweight/obesity in non-pregnant and non-lactating women aged 18–49 years.

Variable	Overweight and Obese			
	No	Yes	X^2^	*p*-Value
VitaminA (μg RE/day)			58.0032	<0.0001
Q1 (<141.77)	1682 (23.24)	1536 (27.26)		
Q2 (141.77~)	1726 (23.85)	1492 (26.48)		
Q3 (267.00~)	1900 (26.25)	1318 (23.39)		
Q4 (485.67~)	1930 (26.66)	1288 (22.86)		
Vitamin B_1_ (mg/day)			2.3649	0.5002
Q_1_ (<0.52)	1840 (25.42)	1378 (24.46)		
Q_2_ (0.52~)	1807 (24.97)	1411 (25.04)		
Q_3_ (0.67~)	1812 (25.03)	1406 (24.96)		
Q_4_ (0.89~)	1779 (24.58)	1439 (25.54)		
Vitamin B_2_ (mg/day)			18.3597	0.0004
Q_1_ (<0.46)	1750 (24.18)	1468 (26.06)		
Q_2_ (0.46~)	1765 (24.39)	1453 (25.79)		
Q_3_ (0.60~)	1819 (25.13)	1399 (24.83)		
Q_4_ (0.77~)	1904 (26.31)	1314 (23.32)		
Niacin (mg NE/day)			91.0130	<0.0001
Q_1_ (<8.72)	1610 (22.24)	1608 (28.54)		
Q_2_ (8.72~)	1764 (24.37)	1454 (25.81)		
Q_3_ (11.88~)	1942 (26.83)	1276 (22.65)		
Q_4_ (15.73~)	1922 (26.55)	1296 (23.00)		
Vitamin C (mg/day)			2.2135	0.5293
Q_1_ (<40.49)	1788 (24.70)	1430 (25.38)		
Q_2_ (40.49~)	1791 (24.74)	1426 (25.31)		
Q_3_ (63.50~)	1820 (25.15)	1399 (24.83)		
Q_4_ (94.29~)	1839 (25.41)	1379 (24.48)		
Vitamin E (mg/day)			31.8191	<0.0001
Q_1_ (<16.48)	1924 (26.58)	1294 (22.97)		
Q_2_ (16.48~)	1831 (25.30)	1387 (24.62)		
Q_3_ (25.12~)	1776 (24.54)	1442 (25.59)		
Q_4_ (37.99~)	1707 (23.58)	1511 (26.82)		
Folate (ug/day)			1.8800	0.5977
Q_1_ (<80.90)	1785 (24.66)	1433 (25.43)		
Q_2_ (80.90~)	1800 (24.87)	1418 (25.17)		
Q_3_ (121.01~)	1816 (25.09)	1402 (24.88)		
Q_4_ (174.09~)	1837 (25.38)	1381 (24.51)		

**Table 4 nutrients-14-01895-t004:** Univariate analysis of the effect of dietary mineral intake on overweight/obesity in non-pregnant and non-lactating women aged 18–49 years.

Variable	Overweight and Obese			
	No	Yes	X^2^	*p*-Value
Calcium (mg/day)			5.7714	0.1233
Q_1_ (<203.17)	1775 (24.52)	1443 (25.61)		
Q_2_ (203.17~)	1787 (24.69)	1431 (25.40)		
Q_3_ (275.62~)	1813 (25.05)	1405 (24.94)		
Q_4_ (378.34~)	1863 (25.74)	1355 (24.05)		
Iron (mg/day)			4.8421	0.1837
Q_1_ (<13.60)	1816 (25.09)	1402 (24.88)		
Q_2_ (13.60~)	1800 (24.87)	1418 (25.17)		
Q_3_ (16.84~)	1768 (24.43)	1450 (25.74)		
Q_4_ (21.35~)	1854 (25.61)	1364 (24.21)		
Zinc (mg/day)			36.6272	<0.0001
Q_1_ (<6.77)	1685 (23.28)	1533 (27.21)		
Q_2_ (6.77~)	1806 (24.95)	1412 (25.06)		
Q_3_ (8.49~)	1822 (25.17)	1396 (24.78)		
Q_4_ (10.56~)	1925 (26.60)	1293 (22.95)		
Potassium (mg/day)			2.3851	0.4964
Q_1_ (<1025.41)	1816 (25.09)	1402 (24.88)		
Q_2_ (1025.41~)	1812 (25.03)	1406 (24.96)		
Q_3_ (1298.68~)	1775 (24.52)	1443 (25.61)		
Q_4_ (1649.00~)	1835 (25.35)	1383 (24.55)		
Sodium (mg/day)			11.4001	0.0097
Q_1_ (<3930.46)	1874 (25.89)	1344 (23.86)		
Q_2_ (3935.97~)	1827 (25.24)	1391 (24.69)		
Q_3_ (5448.84~)	1793 (24.77)	1425 (25.29)		
Q_4_ (7433.89~)	1744 (24.10)	1474 (26.16)		
Selenium (μg/day)			0.5114	0.9164
Q_1_ (<23.80)	1819 (25.13)	1399 (24.83)		
Q_2_ (23.80~)	1798 (24.84)	1420 (25.20)		
Q_3_ (32.62~)	1801 (24.88)	1417 (25.15)		
Q_4_ (44.50~)	1820 (25.15)	1398 (24.81)		

**Table 5 nutrients-14-01895-t005:** Logistic regression analysis of the influence of dietary nutrient intake on overweight/obesity in non-pregnant and non-lactating women aged 18–49 years.

Variable	β	S.E	Wald	*p*-Value	* OR (95% CI)
VitaminA (μg RE/day)					
Q1 (<141.77)					1.00
Q2 (141.77~)	−0.0501	0.0535	0.8763	0.3492	0.951 (0.857, 1.056)
Q3 (267.00~)	−0.2419	0.0568	18.1268	<0.0001	0.785 (0.702, 0.878)
Q4 (485.67~)	−0.2659	0.0619	18.4853	<0.0001	0.766 (0.679,0.865)
Vitamin B_2_ (mg/day)					
Q1 (<0.46)					1.00
Q2 (0.46~)	0.2279	0.0584	15.2018	<0.0001	1.256 (1.120, 1.408)
Q3 (0.60~)	0.3480	0.0677	26.4514	<0.0001	1.416 (1.240, 1.617)
Q4 (0.77~)	0.4156	0.0810	26.3063	<0.0001	1.515 (1.293, 1.776)
Niacin (mg NE/day)					
Q1 (<8.72)					1.00
Q2 (8.72~)	−0.2218	0.0580	14.6098	0.0001	0.801 (0.715, 0.898)
Q3 (11.88~)	−0.4592	0.0673	46.5061	<0.0001	0.632 (0.554, 0.721)
Q4 (15.73~)	−0.4121	0.0782	27.7384	<0.0001	0.662 (0.568, 0.772)
VitaminE (mg/day)					
Q1 (<16.48)					1.00
Q2 (16.48~)	0.1083	0.0524	4.2715	0.0388	1.114 (1.006, 1.235)
Q3 (25.12~)	0.1501	0.0527	8.1064	0.0044	1.162 (1.048, 1.288)
Q4 (37.99~)	0.2106	0.0535	15.483	<0.0001	1.234 (1.112, 1.371)
Zinc (mg/day)					
Q1 (<6.77)					1.00
Q2 (6.77~)	−0.1048	0.0611	2.9449	0.0861	0.900 (0.799, 1.015)
Q3 (8.49~)	−0.0823	0.0736	1.2484	0.2639	0.921 (0.797, 1.064)
Q4 (10.56~)	−0.2413	0.0873	7.6448	0.0057	0.786 (0.662, 0.932)
Sodium (mg/day)					
Q1 (<3930.46)					1.00
Q2 (3930.46~)	0.0350	0.0523	0.4484	0.5031	1.036 (0.935, 1.147)
Q3 (5448.84~)	0.0486	0.0525	0.8562	0.3548	1.050 (0.947, 1.164)
Q4 (7433.89~)	0.0674	0.0529	1.6268	0.2022	1.070 (0.964, 1.187)

* Adjust for age, area types, marital status, education and employment level.

## Data Availability

The data presented in this study are non-public.

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
