# Peer review of "Dietary Micronutrient Status and Relation between Micronutrient Intakes and Overweight and Obesity among Non-Pregnant and Non-Lactating Women Aged 18 to 49 in China"

_nutrients, 2022, doi:10.3390/nu14091895_

Round 1

Reviewer 1 Report

The original article submitted for review assessed the daily supply of micronutrients in the diet of non-preganat and non-lactating women aged 18-49 in China. This article were not present any new information, concludes that the diet of women with normaln and excess body weight is insufficient in terms of the supply of micronutrients (selected vitamins and minerals). Improper diet, especially of people with excess body weight suggests the consumption of highly processed products, which exacerbates the deficiencies described in the work. This information has been known for many years.

Author Response

Thank you for your comments and suggestions. I have added the following content and adjustments in the introduction.

(1) Women between the ages of 18 and 49 are women of reproductive age, whom physical health and nutritional status are not only closely related to successful pregnancy, good preg-nancy outcomes and nurturing of the next generation. Women in perinatal period was focused on supplement micronutrients in China.In social life, non-pregnant and non-lactating Chinese women aged 18-49 are the most vulnerable and neglected group. At present, there is no research data on their dietary micronutrient intake, and the relationship between dietary micronutrient intake and overweight and obesity in China.

(2) Micronutrient deficiencies in non-pregnant and non-lactating women aged 18-49 years can adversely affect fertility, pregnancy outcomes and the risk of congenital disability, harming the health of mother and offspring. However, the non-pregnant and non-lactating women aged 18-49 are the best window of opportunity to implement strategies, correct nutrition and improve physical health.

(3) China Nutrition and Health Surveillance (2015–2017), was the latest cross-sectional survey and was nationally representative. This study aims to use 2015 China Adult Chronic Disease and Nutrition Surveillance (CACDNS 2015) data, a total of 12 872 women aged 18-49 were included in this study,with a large sample size and national representative.

I look forward to your recognition and support. Thanks

Reviewer 2 Report

The authors examined micronutrient intake in non-pregnant and non-lactating Chinese women, and to analyze the relationship between micronutrient intake and overweight and obesity. The intakes of most micronutrients in non-pregnant and non-lactating women were low. The intakes of dietary vitamin A, niacin and zinc were negatively correlated with the risk of obesity, while the intakes of vitamin B2 and vitamin E were positively correlated with the risk of overweight/obesity.

I have following concerns.

  1. Several studies have already described the influence of several nutrient intake on obese people. Authors should clearly describe the originality and novelty in the present study.

  1. This study was to evaluate micronutrient intake in non-pregnant and non-lactating women. Author should more discuss the gender difference in greater detail.

Author Response

1. Thank you for your comments and suggestions. I have added the following content and adjustments in the introduction.

(1) Women between the ages of 18 and 49 are women of reproductive age, whom physical health and nutritional status are not only closely related to successful pregnancy, good preg-nancy outcomes and nurturing of the next generation. Women in perinatal period was focused on supplement micronutrients in China.In social life, non-pregnant and non-lactating Chinese women aged 18-49 are the most vulnerable and neglected group. At present, there is no research data on their dietary micronutrient intake, and the relationship between dietary micronutrient intake and overweight and obesity in China.

(2) Micronutrient deficiencies in non-pregnant and non-lactating women aged 18-49 years can adversely affect fertility, pregnancy outcomes and the risk of congenital disability, harming the health of mother and offspring. However, the non-pregnant and non-lactating women aged 18-49 are the best window of opportunity to implement strategies, correct nutrition and improve physical health.

(3) China Nutrition and Health Surveillance (2015–2017), was the latest cross-sectional survey and was nationally representative. This study aims to use 2015 China Adult Chronic Disease and Nutrition Surveillance (CACDNS 2015) data, a total of 12 872 women aged 18-49 were included in this study,with a large sample size and national representative.

2.  Thank you for your comments and suggestions. I cited some previous studies about the gender difference in my discussion.For instance, The NHANES (2007-2014) also found that dietary zinc intake was negatively correlated with BMI and waist circumference (WC) in women, and the correlation increased with the increase of risk level. The influence of zinc intake on BMI and WC was greater in overweight and obese people, but no such association was found in men. In different studies, vitamin E has a negative or positive relationship with obesity or no effect, and some studies have found that vitamin E intake level is nega-tively correlated with overweight and obesity in women, but not in men. This study did not consider gender difference analysis, and I will consider gender difference in future studies.

I look forward to your recognition and support. Thanks.

Reviewer 3 Report

  • The Introduction Section explains the design of the study. The Authors well justify the research topic. 
  • The study was carried out without methodological errors. 
  • The Descriptions of the results were correct. 
  • The presented figures and table were prepared precisely and also legible. 
  • The Discussion Section included the accurate reference of the results obtained to the studies of other authors.  
  • The Conclusions were well formulated. 

Author Response

Thank you for your comments and suggestions.